



# Detailed detection of fast changes in the active layer using quasi-continuous electrical resistivity tomography (Deception Island, Antarctica)

Mohammad Farzamian[1], Gonçalo Vieira[2], Fernando A. Monteiro Santos[1], Borhan Yaghoobi Tabar[3], Christian Hauck[4], Maria Catarina Paz[1], Ivo Bernando[1], Miguel Ramos[5], and Miguel A. de Pablo[5]

[1] IDL-Universidade de Lisboa, Faculdade de Ciencias, Portugal,
[2] Centre for Geographical Studies, IGOT, Universidade de Lisboa, Portugal,
[3] School of Mining, Petroleum and Geophysics, Shahrood University of Technology, Iran,
[4] Department of Geosciences, University of Fribourg, Switzerland,
[5] Universidad de Alcalá de Henares, Spain

*Correspondence to:* Mohammad Farzamian (mohammadfarzamian@fc.ul.pt)



**Abstract**

Climate induced warming of permafrost soils is a global phenomenon, with regional and site-specific variations, which are not fully understood. In this context, a 2D automated electrical resistivity tomography (A-ERT) system was installed for the first time in Antarctica at Deception Island, associated to the existing Crater Lake site of the Circumpolar Active Layer Monitoring Network (CALM-S) I) to evaluate the feasibility of installing and running autonomous ERT monitoring stations in remote and extreme environments such as Antarctica, II) to monitor subsurface freezing and thawing processes on a daily and seasonal basis and to map the spatial and temporal variability of thaw depth, and III) to study the impact of short-lived extreme meteorological events on active layer dynamics. Measurements were repeated at 4-hour intervals during a full year, enabling the detection of seasonal trends, as well as short-lived resistivity changes reflecting individual meteorological events. The latter is important to distinguish between (1) long-term climatic trends and (2) the impact of anomalous seasons on the ground thermal regime.

Our full-year dataset shows large and fast temporal resistivity changes during the seasonal active layer freezing and thawing and indicates that our system set-up can successfully map spatiotemporal thaw depth variability along the experimental transect at very high temporal resolution. Largest resistivity change took place during the freezing season in April when low temperatures induce an abrupt phase change in the active layer in the absence of a snow cover. The seasonal thawing of the active layer is associated with a slower resistivity decrease during October due to the presence of a snow cover and the corresponding zero-curtain effect. Detailed investigation of the daily resistivity variations reveals several periods with rapid and sharp resistivity changes of the near-surface layers due to the brief surficial refreezing of the active layer in summer or brief thawing of the active layer during winter as a consequence of short-lived meteorological extreme events. These results emphasize the significance of the continuous A-ERT monitoring set-up which enables to detect fast changes in the active layer during short-lived extreme meteorological events.

Based on this first complete year-round A-ERT monitoring data set in Deception Island, we believe that this system shows high potential for autonomous applications in remote and harsh polar environments such as Antarctica.



# 1 Introduction

Although permafrost soils show currently a clear global warming trend due to climate change (Biskaborn et al. 2019), regional differences can be pronounced, which are not only due to regional climate differences but also due to heterogeneous soil characteristics. One example is the Antarctic Peninsula where one of the strongest air temperature increases is recorded

since the 1950's. In spite of this general air temperature increase, the northwest of the Antarctic Peninsula has shown a cooling trend between 1999 and 2015 (Turner et al., 2016; Oliva et al., 2016). Consequently, and contrary to the general trend, the seasonal surficial thaw layer of the ground above the permafrost (the active layer) decreased, indicating that the climate signal is more complex than previously accounted (Ramos et al. 2017).

The active layer of permafrost environments is not only a climate change indicator, but  its dynamic is of extreme importance

to terrestrial ecosystems since it influences the hydrology, soil nutrient and contaminant fluxes, as well as geomorphological processes, such as soil erosion and mass wasting. Furthermore, changes in active layer thickness may also affect infrastructure due to the effects it shows on the rheological properties of the perennially frozen soil (Williams and Smith, 1989).

In moist Polar environments, the transition zone between the active layer and the permafrost table is frequently characterized

by the presence of a high content of interstitial ice, forming an ice-rich layer, some centimeters to decimeter thick. This relates to the interannual variability of the active layer thickness. In warmer summers, the active layer thickens and water percolates downwards concentrating at the permafrost table, refreezing at the beginning of the cold season. In cooler summers, the active layer is shallower and the previously formed ice does not melt. This ice-rich layer, still poorly characterized, but with significance due to its impacts on soil behavior is called the transient layer (Shur et al., 2005). A

continuous monitoring of the physical properties of the active and transient layers is therefore essential to understand the permafrost dynamics and its potential impacts on climate feedbacks and local ecology.

Deception Island in the South Shetlands archipelago, off northern Antarctic Peninsula, is an extraordinary natural laboratory to study active layer and permafrost dynamics. The island is an active stratovolcano with widespread permafrost down to sea-level except at spatially restricted localities with geothermal anomalies, generally along faults (Goyanes et al., 2014). The soil

surface is bare with vegetation almost completely absent, and permafrost is close to its climatic limit since mean annual air temperatures are just below 0 ℃ (Bockheim et al., 2013; Ramos et al., 2017). The soil is composed by a mix of lavas, lapilli, and pumice, which in some areas induce high thermal insulation, with resulting active layer thickness of only 40 cm.

The shallow active layer and soil characteristics of Deception Island, the easy access to the permafrost table, as well as the geographical setting in the Maritime Antarctic and its geothermal characteristics, have made the island one of the best-studied

areas for permafrost research in the Antarctic Peninsula (e.g. Ramos et al., 2008; Vieira et al., 2010; Bockheim et al., 2013; Melo et al., 2012; Goyanes et al., 2014; Ramos et al., 2017). Two permafrost and active layer monitoring sites within the Circumpolar Active Layer Monitoring – South Program (CALM-S) and the Global Terrestrial Network for Permafrost (GTN-O/GCOS/IPA) including ground temperature boreholes and meteorological stations have been installed, at Irizar col and Crater Lake.



So far, monitoring of the active layer dynamics in Antarctica was conducted using only 1-dimensional borehole and meteorological data, which restricted the analysis to point information that often lack representativeness at the field scale. In addition, being an invasive technique, the drilling of boreholes disturbs the subsurface and is not feasible to conduct over large areas, especially in environmentally sensitive ecosystems such as the Antarctic.

As a cost-effective and ecologically non-hazardous alternative, 2-dimensional geophysical monitoring, such as Electrical Resistivity Tomography (ERT), allows for monitoring the spatiotemporal variability of the freezing and thawing characteristics of the active layer and the permafrost, as has been demonstrated in several applications in the European Alps (e.g. Hauck, 2002; Hilbich et al., 2008; 2011; Krautblatter et al., 2010; Ottowitz et al., 2011; Supper et al., 2014; Mewes et al., 2017). ERT is a non-invasive technique that is sensitive to the electrical conductivity (the reciprocal of electrical

resistivity) of materials. Due to the large contrast between the resistivity of ice and water, the method has become popular in permafrost investigation to distinguish between frozen and unfrozen soil.

Although individual ERT measurements in Antarctica have been reported (e.g. McGinnis et al., 1973; Guglielmin et al., 1997; Gugliemin and Dramis, 1999; Hauck et al., 2007; Goyanes et al., 2014), no continuous and autonomous ERT monitoring has been attempted yet at these remote and extreme environments, where winter access is usually impossible. In

these cases, maintenance and repair, which has often become necessary in the autonomous ERT studies reported from the European Alps (cf. Supper et al. 2014), is not possible for most of the year. In this paper, we show that continuous ERT monitoring of the active layer and shallow permafrost is possible in Antarctica, and that its results may yield high-resolution 2-dimensional data on freeze and thaw characteristics on different time-scales.

We installed and tested an autonomous and continuously measuring ERT monitoring system in the vicinity of shallow

boreholes at the Crater Lake CALM-S site, Deception Island with the objective to evaluate its potential in a remote area without maintenance for a full year. The Crater Lake CALM-S site is typical for conditions found in Antarctica, where year-round stations are scarce, and most research stations are only summer operated. Data were collected to monitor subsurface freezing and thawing processes on a daily and seasonal basis, and to detect seasonal trends as well as the impact of short-lived extreme meteorological events. Short-lived meteorological events are rarely addressed in permafrost studies, but they

reflect the impact of fast-changing meteorological conditions on the upper soil horizons. In the context of climate change, with increasing frequency of extreme events, these events may also become more frequent. Being able to identify them in the ERT series allows for a better characterization of the links between soil thermal regimes and geomorphic dynamics.

## 2 Study area

Deception Island (62° 55′ S, 60° 37′W) is located about 100 km north of the Antarctic Peninsula (AP), in the Bransfield Strait and is part of the South Shetlands archipelago (Fig. 1). The island is a stratovolcano with a horseshoe shape and a diameter of 15 km, a 7 km wide caldera open to the sea and maximum elevation at Mount Pond (539 m). About 57% of the island is currently glaciated and about 47 km$^2$ are glacier-free (Smellie and López- Martínez, 2002). The climate is cold-oceanic with frequent summer rainfalls, a moderate annual temperature range and mean annual air temperatures close to -3 °C at sea level.



The weather conditions are dominated by the influence of the polar frontal systems and atmospheric circulation is very variable including the possibility for winter rainfall (Styszynska, 2004). Deception Island is an active volcano and is formed by intercalation of lava flows, pyroclastic and ash deposits, with many of the present-day glaciers ash-covered. During the recent eruptions of 1967, 1969 and 1970, pyroclastic and ash deposits covered the snow mantle, and buried snow is still present at some sites. Deposits are very porous and insulating with high ice content at the permafrost table. The active layer is thin, varying from 30 to 96 cm depth across Deception Islands in different soils (Bockheim et al., 2013) and boreholes show the presence of warm permafrost.

The Crater Lake CALM-S site is located in a small and relatively flat plateau-like surface covered by volcanic and pyroclastic deposits at 85 m a.s.l, north of Crater Lake (62°59′06.7″ S, 60°40′44.8″ W). The site was selected due to its flat characteristics, absent summer snow cover, large distance to known geothermal anomalies, good exposure to the regional climate conditions (mitigating site-specific effects and being representative in a regional context) and because of the vicinity to the Spanish station Gabriel de Castilla. The ground surface is completely devoid of vegetation and the MAAT at the Crater Lake CALM-S site between 28/01/2009 to 22/01/2014 was −3.0 °C. Permafrost temperatures are -0.3 °C to -0.9 °C, with permafrost thickness varying spatially from 2.5 to 5.0 m (Vieira et al., 2008; Ramos et al., 2017) and active layer thickness in the range of 25 to 40 cm. This spatial variability has not been addressed in the literature, but it is possibly related to differences in surface deposits and snow cover.

Figure 1 (near here)

## 3 Material and Methods

### 3.1 Crater Lake CALM-S environmental monitoring setup

The Crater Lake CALM-S site consists of a 100 × 100 m grid with 121 nodes spaced at 10 m intervals and was installed in January 2006 (Fig. 2) with several upgrades since then. The site includes monitoring of air temperature, permafrost and the active layer in boreholes, snow thickness and once per year, thaw depth is measured manually by mechanical probing during the summer (Ramos et al., 2017). The topography map of the CALM-S site shows relatively a flat area with a maximum of 6 m variation in elevation within the site.

Air temperatures are measured at 160 cm above the surface, monitored with hourly measurements since 2009. Ground temperatures are measured in the shallow borehole $S_{3,3}$ down to 160 cm (node 3,3) (Fig. 2). This borehole has a diameter of 32 mm and is cased with air-filled PVC pipes and ground temperatures are measured with ibutton-sensors at depths 2.5, 5, 10, 20, 40, 80 and 160 cm. In addition, ground temperatures are measured in 16 very shallow boreholes regularly distributed within the grid, with a single ibutton sensor close to the base of the active layer. Finally, snow thickness is estimated using a series of near-surface air temperature ibutton miniloggers installed in a vertical stake at 5, 10, 20, 40, 80 cm height above the ground (de Pablo et al., 2016). Snow distribution is mapped using a Campbell CC640 time-lapse camera with daily pictures



at 11:00, 12:00 and 13:00 (local solar time). The combined approach of snow pole and the time-lapse camera allows evaluating the snow distribution in the study area.

Figure 2 (near here)

### 3.2 Electrical Resistivity Tomography monitoring

Electrical Resistivity Tomography (ERT) is the method for the calculation of the subsurface electrical resistivity distribution from multiple electrical resistance measurements made using a quadrupole arrangement of electrodes. The electrodes are placed on the ground surface and a 2-D or 3-D image of the resistivity distribution can be achieved by varying the location

and spacing of the electrodes. The relationship between the measured spatial apparent resistivity distribution and the true resistivity distribution of the subsurface is complex and needs to be estimated using inversion theory (Loke, 2002). Under the assumption that general conditions (e.g. lithology, pore space) remain unchanged during the observation period, repeated resistivity measurement can provide a mean for evaluation of freezing or thawing processes and subsurface temperature variations (Hauck, 2002).

An automatic ERT (A-ERT) monitoring system using a 4POINTLIGHT_10W (Lippmann) instrument, was installed in the vicinity of the ground temperature borehole $S_{3,3}$ (see Fig. 2) in 2010 in order to monitor active layer freezing and thawing by ground surface time-lapse surveys (Fig. 3). The system was installed close to the interfluve, in the most elevated zone within the site, where stronger spatiotemporal subsurface variations are expected. The Lippmann resistivity meter was programmed in combination with multi-electrodes for ERT surveys. The readings were converted to apparent resistivity values and stored

in the internal memory.

All ERT surveys were performed using the Wenner electrode configuration to minimize energy consumption and measurement time as well as to obtain the best signal-to-noise ratio in highly resistive terrain (Kneisel, 2006; Hauck and Vonder Mühll, 2003). The Wenner array is also more sensitive to vertical changes in the subsurface resistivity below the center of the array (Loke, 2002) which makes the configuration ideal for active layer imaging. 20 copper plates, which are

connected by buried cables to the active boxes, with an electrode spacing of 0.5 m were used in this study (Fig. 3b). A robust, water-proof box was used and buried, casing the 4POINTLIGHT_10W instrument, solar panel-driven battery and multi-electrodes connectors during data acquisition (Fig. 3c). This setup yields 56 individual data points for each monitoring data set at six data levels (Fig. 3d). A-ERT measurements were started at the beginning of 2010 and repeated every four hours during one full year.


Figure 3 (near here)





### 3.3 A-ERT data processing and inversion

A-ERT data processing and inversion include several steps: data filtering (outlier detection), spatial mean apparent resistivity analysis and resistivity data inversion. As a first step, the apparent resistivity data measured during one year were filtered by removing data spikes and negative values. Furthermore, data points with standard deviations of more than 2% after 9 stackings were excluded. The quality of the apparent resistivity data was good in most datasets and only less than 0.5% of all measurements had to be eliminated. The measured apparent resistivity data were then averaged for each depth level, providing six horizontal mean values for each dataset as shown in Fig. 3d, and analyzed regarding daily and monthly resistivity changes.

In the next step, the apparent resistivity datasets were inverted using e.g. the commercially available software RES2DINV. The robust inversion option in RES2DINV, as well as a mesh refinement to half of the electrode spacing, was applied to better resolve the expected strong resistivity contrasts between unfrozen and frozen subsurface materials. In addition, a full 4D inversion algorithm developed by Kim et al. (2009) was used in this study to better image the temporal resistivity changes. The full 4D inversion algorithm defines a subsurface structure and the entire monitoring data in the space-time domain to obtain a four-dimensional space-time model using just one inversion process. In this approach, regularizations are introduced not only in the space domain but also in time, resulting in reduced inversion artifacts and improved stability of the inverse problem (Kim et al., 2009).

### 3.4 Virtual borehole analysis

A so-called virtual borehole analysis (e.g. Hilbich et al., 2011) was used to further investigate the dynamics of the active layer during 2010 in more detail in order to evaluate the temporal variation of the thaw depth as well as to study the resistivity-temperature relationship. Here, inverted resistivity values were extracted from the tomogram along a 1-dimensional depth transect, close to the existing borehole $S_{3,3}$. At this borehole, temperature sensors are installed at different depths down to 160 cm, and temperature data are available every 3 hours during the experiment. The maximum depth of the ERT investigation is hereby almost equal to the deepest temperature sensor. The inverted temporal vertical resistivity variations from the tomogram are then compared to the corresponding temporal thermal variations obtained from $S_{3,3}$.

## 4 Results and Discussions

### 4.1 Analysis of observational data

Figure 4 shows air, surface, shallow and ground temperature variations during the A-ERT monitoring period observed very close to the A-ERT transect. Snow cover during winter is thin with only 10 to 20 cm thickness and frequent snow-free periods. Correspondingly, air and ground temperature are generally well-coupled with a slight phase lag in the presence of snow (cf. the cooling events in August and September 2010).



The ground temperature at $S_{3,3}$ at shallow depths (Fig. 4b) fluctuates significantly during the year, with temperature ranging from −8 to 5 °C, reflecting the snow cover variability and air temperatures. Temperatures above zero are delineated by the yellow to red colors, indicating the active layer thawing events. The temperature of the active layer falls below 0 ℃ at the end of April and stays below 0 ℃ until the beginning of November. The zero-curtain phase in spring is around 1 month, whereas no significant zero-curtain can be seen in autumn due to the low air and soil surface temperatures and the absence of a thick snow cover during freezing. Short-lived meteorological events with quick and superficial changes of the ground temperature around 0 ℃ are quite frequent during the study period, and therefore, brief surficial refreezing (e.g. in March, April, and December) and thawing of the active layer (May) can be identified in the summer and winter respectively.

The temperature variations of the shallow temperature boreholes at nodes 2,2 and 4,2 (see Fig. 2 for the locations of the temperature sensors) are shown in Fig. 4c to investigate the lateral temperature changes along the A-ERT transect. Node 2,2 is closer to the interfluve and is more wind and sun exposed. Consequently, a thinner snow cover, as well as smaller number of days with snow, were recorded at node 2,2 when compared to node 4,2 with corresponding lower winter temperatures at node 2,2 due to the stronger insulation effect of snow at node 4,2 (supported by time-lapse camera observations). The stability of temperatures at 0 ℃ in the summer months reflects the ice-content and latent heat effects that limit thaw propagation.

Figure 5 shows the spatial distribution of thaw depth across the Crater Lake CALM-S site, measured in January 2010. The thaw depth varies between 25 to 40 cm with shallower thaw depth in the south of the study area where the area is less wind-exposed and shows a longer and more stable snow cover. The thaw depth is approximately 30 cm along the A-ERT transect in January 2010.

Figure 4 (near here)

Figure 5 (near here)

## 4.2 Apparent resistivity data

The apparent resistivity raw data of all surveys were processed as discussed in Sect. 3.3 and the resulting mean daily apparent resistivity change of each data level are shown in Fig. 6a. The usefulness of investigating spatiotemporal apparent resistivity data over different timescales was demonstrated in several studies (e.g. Hilbich et al., 2008; 2011). They allow insights into the resistivity variability trend during the year as well as the identification of the impact of specific meteorological events on the subsurface thermal regime. For most of the year, resistivity increase and decrease can be associated with freezing and thawing processes.





The apparent resistivity data collected at a=1 and 2 levels (corresponding to 0.5 and 1 m electrode spacing respectively) reveal a sharp resistivity rise on April, 19th from approximately 10-20 kΩ.m and reaching to values more than 500 kΩ.m on May 5, suggesting the beginning of the seasonal freezing of the active layer. Because of the absence of a snow cover during this period, the very low air temperature provokes an abrupt phase change which causes a sharp resistivity rise in this period.

The delayed response of deeper levels (i.e. a=3, 4, 5 and 6; corresponding to 1.5, 2, 2.5 and 3 m electrode spacing respectively) indicates the advancing freezing front and is coincident with the gradual decrease of the active layer temperature with depth (see Fig. 4b). The freezing of the active layer intensifies in June, July, and August. The beginning of the seasonal thawing phase is associated with the steady decrease of apparent resistivity, starting on October 4th from a value of approximately 200 kΩ.m, to less than 40 kΩ.m at the end of October. During the seasonal thawing of the active layer, the

snow cover dampens the thawing effect and provides water input to the active layer, which refreezes again at the still frozen active layer. Interestingly, this zero-curtain phase, visible in the temperature record, was reflected in the steady decrease of apparent resistivity, recorded by the A-ERT system in this period. Deeper levels experience the resistivity decrease with some delay.

In general, the daily apparent resistivity fluctuations are relatively small. However, Fig. 6a reveals several significant

resistivity fluctuations during the observation period. These fluctuations are associated with either brief surficial refreezing of near-surface layers in summer, or short thawing periods during winter as a consequence of short-lived meteorological extreme events with quick and superficial changes of the ground temperature around 0 ºC. Two examples of these daily apparent resistivity changes during the short-lived events, events (I) and (II), were selected for detailed investigation. Event (I), shown in Fig. 6b is an example of the surficial refreezing of the active layer in the summer. A continuous increase of

apparent resistivity at shallower levels is evident with a total difference of approximately 30 kΩ.m in 10 days. On the other hand, Event (II), shown in Fig. 6c, presents a very rapid apparent resistivity decrease at shallowest levels, a=1 and a=2, with a total difference of approximately 400-600 kΩ.m in 3 days. The observation of such rapid changes of the apparent resistivity proves the significance of the automatic ERT monitoring system to record continuous resistivity changes.

Figure 6 (near here)

### 4.3 Monthly resistivity variations

A monthly selection of the modeled resistivity data in 2010 and the resistivity changes relative to the first ERT dataset are shown in Fig. 7. Data collected on the 28th day of each month at 12:00 was used in this analysis and all data was inverted

using the full 4D inversion algorithm, described in Sect. 3.3. The corresponding temperature profiles were marked with the dashed lines in Fig. 4b.

The resistivity pattern along the A-ERT monitoring transect at CALM-S site is characterized by two vertical distinct resistivity zones. The first zone, down to 20-40 cm depth in summer images the active layer. The resistivity of this layer



changes dramatically during freezing and thawing. The deeper zone images the permafrost to a depth of 160 cm during the A-ERT measurements. The resistivity of both active layer and permafrost zones indicate only a slight lateral change along the transect. This can be well explained by the spatial homogeneity of the study area, as well as by the small size of the A-ERT transect, which is smaller than for other A-ERT studies in which stronger lateral variations along the ERT transects are

usually more evident (i.e. Hilbich et al., 2011; Supper et al., 2014; Keuschnig et al., 2017) .

The resistivity model plotted for January shows a more conductive zone (less than 10 kΩ.m) for the first 30-40 cm, followed by a deeper zone with resistivities of more than 30 kΩ.m. The shallow zone images the active layer in summer when this layer has not been frozen yet and shows a slight thickness increase from the left to the right. The thickness and small lateral variability of this layer are in good agreement with the thaw depth measurement using a mechanical probe in January 2010

(cf. Fig. 5). The resistivity and thickness of the active layer show a slight change during February and March. However, a more significant resistivity decrease is evident at depths of more than 50 cm due to the slight temperature increase at depth during February and March (Fig. 4b) which increases the unfrozen water content and consequently decreases the subsurface resistivity.

The largest resistivity changes at the surface during the year take place between March and April due to the freezing of the

active layer. Interestingly, the resistivity of the permafrost at more than 1 m depth decreases slightly in this day. The resistivity model behavior in April can be well explained with an abrupt phase change during the active layer freezing in shallow surface and delayed response of the deeper zone. This is in very good agreement with the thermal transect shown in Fig. 4b, which shows a slight temperature increase at more than 1 m depth when compared to March. On the other hand, the resistivity model plotted for May is characterized by a resistivity decrease at the shallower zone (active layer) and a resistivity

increase at depth (permafrost) when compared to the resistivity model plotted for April. The resistivity increase of the permafrost is coincident with a temperature decrease of the permafrost during May which results in lower unfrozen water content. On the other hand, the active layer warming during this month provides more unfrozen water to the active layer which decreases the active layer resistivity during the same period.

The freezing of the active layer and cooling of permafrost intensifies during June, July, and August which is reflected by the

high resistivity values and then the beginning of seasonal thawing is associated with the resistivity decrease in October. The average resistivity of the active layer on 28[th] October is higher than the corresponding zone during the thawing seasons (i.e. January, February, March, November, and December). This is due to the zero-curtain phase in October when the snow cover damps the thawing effect and the temperature stays around zero. The steady resistivity decrease of the active layer and permafrost until a depth of 160 cm is evident along the A-ERT transect during November and December due to the

subsurface temperature increase. The monthly subsurface resistivity behavior is consistent with the mean apparent resistivity data shown in Fig. 6a.

Figure 7 (near here)



### 4.4 Daily resistivity variations on the scales of individual events: Events (I) and (II)

Two short-lived meteorological events with fast and superficial changes of the ground temperature around 0 ℃ are selected for detailed A-ERT analysis to investigate how well the A-ERT model can resolve the expected sharp subsurface changes associated with the fast active layer freezing and thawing processes.

Figure 8a and 8b show in detail the air and ground temperature fluctuation during the selected events. Event (I) indicates a surficial refreezing of the active layer in the summer. A decrease in air temperature started on March 16 and intensified on March 20 with a subsequent increase starting from March 22. The arrival of the cold air induced an impact in the uppermost 5 cm starting from March 17, when the ground temperature at depths 2.5 and 5 cm falls below zero. The active layer refreezing intensified between 22$^{nd}$ and 24$^{th}$ March when temperatures decreased and the advancing freezing front reached 10

cm. A very shallow subsurface phase change is expected during this short-lived meteorological event as no impact has been recorded at ground temperature sensors deeper than 10 cm.

Event (II) presents a surficial thawing of the active layer in summer. A drastic rise in the air temperature from -8.4 to 1.4 ℃ is evident on May 9. The warm air influenced the ground temperature immediately and generated an abrupt phase change in the top 20 cm on May 10, as evidenced by the above-zero temperatures on May 10 and 11 at depths 2.5, 5, 10 and 20 cm. The

time-lapse camera photos taken on May 9 (Fig. 8c) show clearly the fast snow melt between 11h00 and 13h00 on this day, which might explain the quick subsurface temperature rise due to the infiltration of the melted snow to the soil subsurface and consequent advective heat transfer. The thermal sensors at depths 40 and 80 cm also recorded the temperature increase during this event although the temperature stays below zero at these depths. Event (II) lasts for a shorter period compared to the event (I). However, it caused stronger and deeper subsurface temperature changes.

Figure 9 shows the time-lapse inversion results during the events (I) and (II). Data collected on March 14 and May 7 were used as the reference for the events (I) and (II) respectively, and the resistivity changes relative to the first ERT dataset are presented in this figure. Data collected at 12h00 was used in this analysis and all data was inverted using the full 4D inversion algorithm, discussed in Sect. 3.3. A continuous resistivity increase at shallow depth (less than 30 cm) is evident from March 18 till March 24 in Fig. 9a. The resistivity of this zone started to decrease again on March 26, and there is no

significant change between resistivity models on March 28 and the reference model on March 14. In addition, no significant change occurs at depths of more than 30 cm during this event, suggesting that this event provoked phase changes only within the shallow subsurface. The results of the time-lapse resistivity models are in good agreement with the air and ground temperature fluctuation shown in Fig. 8a. The resistivity increase of the active layer is coincident with the temperature decrease of the active layer at shallow depth. The resistivity of the active layer reached its maximum between March 22 and

24 when the temperature reached its minimum and a larger amount of the pore water is frozen.

Figure 9b shows a sharp resistivity decrease on May 9 suggesting an abrupt phase change during this day. In the following, the resistivity of the active layer reached its minimum on May 10 and 11. We anticipate that this is due to the infiltration of the snow-melt water into the soil subsurface which provides liquid water to the active layer and decreases resistivity. A slight increase of resistivity at depths of more than 1 m is evident on May 9 and 10. This can be explained by the slight permafrost

temperature decrease at a depth of 160 cm on these days (cf. Fig. 8b). Interestingly, the resistivity models suggest the



propagation of the thawing process from the left (A) to the right (B) on May 9 and 10 (i.e. the active layer resistivity decreased from the left to the right) and then refreezing from the same direction on May 11. Because the left side of the A-ERT transect is closer to the interfluve and is more wind and sun exposed, subsurface thaw and snowmelt are expected to take place from left to right along the transect orientation after the initial air temperature rise on May 9. Similarly, active

layer refreezing starts from the same direction (left to right) when the air cools down again on May 11. The continuous active layer refreezing during the following three days is coincident with a slight resistivity decrease at depth. This can be explained with the delayed response of the permafrost to the temperature signal at the surface. An increase of the permafrost temperature at depths 40 and 80 cm was recorded in the ground thermal sensors on these days.

Figure 8 (near here)

Figure 9 (near here)

### 4.5 Temperature-Resistivity relationship

The temperature-resistivity relationship in temperatures below zero was studied during two periods: I) the beginning of the seasonal active layer freezing in April/May (P1) and II) the beginning of the seasonal thawing in October (P2). The selected A-ERT data was inverted using the full 4D inversion algorithm, described in Sect. 3.3. The virtual borehole analysis, described in Sect. 3.4 was used to establish the temperature-resistivity relationship.

Figure 10 shows the linear regression between the resistivity and temperature in the virtual borehole at the $S_{3,3}$ at three depths
(i.e. 20, 40 and 80 cm). These depths were selected to study the resistivity – temperature behavior of the active layer (20 cm), permafrost table (40 cm) and permafrost (80 cm). The figure shows an excellent linear regression between the resistivity and temperature at all depths during the seasonal thawing in October with $R^2$ greater than 0.96. Small deviations from this linear relationship can be found during the seasonal freezing phase due to the faster subsurface temperature and therefore phase change (no zero-curtain present), where the mismatch between the volume measured by resistivity and temperature can be
larger. In addition, the effect of downward ion migration upon freezing may further influence the relationship; however, a strong linear regression between the resistivity and temperature at all depths can be also seen ($R^2$ greater than 0.86) in all depths in this period. Highest resistivities (and probably ice contents) are found at 40 cm depth upon freezing, which may partly be due to more frequent refreezing effects at the boundary between active layer and permafrost.

Figure 10 (near here)



**4.6 Evaluation of the temporal resistivity variability in the virtual borehole S₃,₃**

Figure 11 shows the resistivity evolution with time in virtual boreholes at the $S_{3,3}$ location during 2010 using the evaluation described in Sect. 3.4. As the RES2DINV software cannot invert more than 21 ERT datasets simultaneously, no time-lapse inversion algorithm was used for this analysis and apparent resistivity data were inverted independently using a batch routine.

The zero degree isotherm from the borehole temperatures at $S_{3,3}$ (Fig. 4b), is superimposed on the resistivity tomogram. A resistivity cut-off value of 13 kΩ.m was selected to delineate the temporal variability of the thaw depth at the $S_{3,3}$ location. This value was selected based on our analysis of the individual resistivity tomograms as well as the average thaw depth measured by a mechanical probe in January. This value roughly corresponds to the resistivity transition value between the unfrozen media at the surface and the more resistive frozen zone at depth.

The average of thaw depth at the end of January is about 30 cm with a slight increase in February. The brief active layer thinning between 20 and 24 of February might have happened due to the brief active layer cooling in this period, recorded by the ground temperature sensors. Afterwards, the thaw depth increases to an average of 40 cm at the beginning of March. The maximum thaw depth is recorded during March probably due to the stronger active layer warming in this month. The sudden resistivity rise in the middle of March is coincident with the brief active layer freezing (event I), discussed in Sect. 4.4.

Thinning of the active layer starts in April due to the active layer cooling and possible refreezing of the infiltrating water above the permafrost table.

The largest resistivity changes in the active layer took place at the end of April due to the active layer freezing. The active layer stays frozen from May until October except during the brief surficial thawing event between 7 and 14 of May (cf. Fig. 9b). The resistivity changes near the surface during the winter are coincident with consecutive active layer cooling and

warming. The resistivity values are greatest in winter and around the permafrost table at depths around 40 cm. We anticipate that this is due to the repeated thawing and refreezing processes of water infiltrating from snow/rain that accumulated on top of the permafrost table (cf. the critical zone, Shur et al., 2005)

During the zero-curtain phase in October-November, the ground temperatures are still below zero and the active layer is still frozen. However, unfrozen moisture is already present due to snowmelt and the warm but subzero temperatures, which

results in lower resistivity values near the surface. The active layer thaws at the beginning of November when the temperature rises above zero and is coincident with the strong resistivity decrease in this period. The average thaw depth in November-December is 20 cm with a slight increase at the end of December. The sudden resistivity rise in December is coincident with the brief active layer freezing in this month.


Figure 11 (near here)



## 5 Discussion

The monitoring setup with a very small electrode spacing (i.e. 50 cm) and dense measurements of six times per day was designed to generate subsurface resistivity maps with very high spatial and temporal resolution. This enables to detect the expected fast and sharp resistivity changes within the very narrow active layer during the short-lived extreme meteorological events at the study site. Since short-lived meteorological events may induce phase change, they are potential generators of geomorphic activity, such as cryoturbation, or even small debris-flows in sloping terrains. These events are particularly important in regions without a thick or continuous snow cover such as the Deception Island due to the quick response of the active layer to the air temperature signal.

With this high-resolution set-up, we were able to identify these events in our A-ERT models. Looking more closely at the resistivity and temperature changes during the brief active-layer thawing events, we suggest that infiltration processes from the melting snow cover are the dominating factor provoking the observed resistivity decrease and temperature increase. This is in agreement with Scherler et al. (2010) who simulated the active layer thaw period using a 1-dimensional fully coupled heat and mass transfer model. They found that the water pool, formed at the ground surface from the melting snow cover, may percolate and reach greater depths which results in fast water and advective heat transfer to depth. The infiltration ends when the water pool is emptied and/or the water refreezes. Such shallow active layer dynamics show that there are freeze-thaw cycles during the freezing season, which may result in cryoturbation and in sloping terrain, be responsible for increased superficial solifluction.

We used a resistivity cut-off value of 13 kΩ.m to distinguish the unfrozen media at the surface (i.e. active layer) from the more resistive frozen zone (i.e. permafrost) at depth. This value was selected empirically based on our analysis of the individual resistivity tomograms and the average thaw depth measured by a mechanical probe in January. The resistivity of the unfrozen media in the study area is very high compared to other studies conducted in alpine and polar regions (e.g. Supper et al. 2014, Keating et al. 2018), and could be due to the soil being composed by very porous lapilli with high air content and large intergranular pore spaces that induce fast percolation of snow melt and rainwater. The dark surface of the soil and the wind-exposed conditions promote fast evaporation and favors a quick drying of the near-surface horizons. It is worth mentioning that the resistivity of any subsurface material is a complex function of soil properties (e.g. grain and pore size, void ratio, degree of saturation, water content and salinity, temperature and water phase) and thus the cut-off value cannot be used for other sites and a site-specific investigation is required to estimate this value.

The detailed investigation of the resistivity tomograms indicates that our A-ERT set-up could better map the thaw depth, compared to the ground temperature sensor $S_{3,3}$. In fact, the thaw depth variability in the 10, 20 and 40 cm depth range, seen in the resistivity tomogram, is not reflected in the borehole temperature data due to the lack of sensors between these depths. Hence, the ground temperature tomogram (Fig. 4b) shows a constant thaw depth of 40 cm in the first three months. On the other hand, our resistivity models slightly overestimated the thaw depth in several periods, when compared to the borehole temperature data. Examples of such overestimations are seen in March when inverted resistivity suggests the thaw depth to be slightly over 40 cm. This error becomes worse at the beginning of the seasonal thawing in November; when the A-ERT derived thaw depth is too thin (5 cm).




We used a small electrode spacing of 50 cm to deal with the expected abrupt changes near the surface. However, the resolution of the A-ERT within the first 10 cm (e.g. the active layer condition at the beginning of the seasonal thawing) is very limited. In addition, the over-parameterized inverse problem and the effects of smoothing from regularization applied in the inversion algorithm overestimate the thaw depth in the resistivity tomograms. In the virtual borehole analysis, each dataset was inverted independently and temporal resistivity changes of individual quadrupoles were not accounted for in the inversion. Using a time-lapse inversion algorithm as used in Sect. 4.3 and Sect. 4.4 might enhance the temporal resolution of the resistivity tomogram and reduce the uncertainty in the estimation of the thaw depth.

In accordance with other studies (e.g. Oldenborger and LeBlanc, 2018) the obtained linear relationship between resistivity and temperature implies the absence of large latent heat effects during phase change, i.e. comparatively dry conditions. In addition, in pyroclastic sediments, pores containing water can be disconnected from each other, which further reduce the effect of phase change between liquid and frozen water on the bulk resistivity.

## 6 Conclusion and outlook

An automated ERT (A-ERT) system with a solar panel-driven battery and multi-electrodes configuration was installed at Deception Island at the Crater Lake CALM-S monitoring site, as the first automatic resistivity monitoring system in Antarctica. Our analysis of this combined geophysical and thermal monitoring approach focused on (I) the ability of the A-ERT system to monitor the spatiotemporal variability of the active layer along the small-scale transect (II) the active layer freezing and thawing processes on seasonal time scales and (III) the impact of extreme short-lived meteorological events on the ground thermal regime.

Based on the comprehensive analysis of the A-ERT data, the following main conclusions can be drawn:

1) The A-ERT system allows monitoring in detail the spatio-temporal variability of the active layer in summer. The maximum thaw depth in 2010 was recorded in March with values slightly more than 40 cm.

2) The process of active layer freezing in autumn and thawing in spring was well resolved by the A-ERT system. The absence of the snow cover and direct influence of atmospheric processes during the seasonal freezing provoked a drastic resistivity rise in April. On the contrary, the zero-curtain phase during the seasonal thawing causes a continuous resistivity decrease during several weeks in November.

3) Short-lived meteorological events during a few days provoked a fast and dramatic resistivity change in the active layer due to the brief active layer freezing and thawing, detected by the A-ERT system. Our study clearly shows that without automatic and quasi-continuous measurements, short-time active layer freezing and thawing, as well as the infiltrating water from the melting snow cover to the ground during such extreme meteorological events, could not be investigated.

The automated system developed in this study allows a free choice of measurement interval as well as electrode configuration, and our A-ERT set-up with a small electrode spacing of 0.5 m and dense measurements of six times per day



enabled us to detect the impact of the extreme short-lived meteorological events on the active layer with a thickness as small as of 20-40 cm. Interestingly, our A-ERT system could detect the spatial directions of the thawing and freezing processes along such a small transect.

The consistency of our full year results with previous studies in more easily accessible alpine and polar regions (e.g. Hilbich

et al., 2011; Supper et al., 2014; Keuschnig et al., 2017; Tomaskovicova, 2017, Oldenborger and LeBlanc, 2018) suggests that the detailed studies of the Alps can be transferred to set-ups in very remote environments, which would allow for integrative process studies as well as coupled modeling of A-ERT data with existing water content and temperature monitoring system in Antarctica. Examples of such joint geophysical and thermal modelling approaches were given in Scherler et al. (2010) using uncoupled models and Tomaskovicova (2017) using a fully coupled electro-thermal modelling

approach.

A long-term deployment of an A-ERT system in Antarctica would allow a much more detailed analysis of the permafrost and active layer evolution, which could be used as input data for hydro-thermal models simulating the future permafrost evolution (e.g. Marmy et al., 2016, Rasmussen et al., 2018). On a more local scale, the specific characteristics of Deception Island, where permafrost conditions are influenced also by geothermal and even volcanic activity, would allow for detailed

investigations of the resulting hydro-thermal interactions in a cryospheric context. The fact that the monitoring occurs along a transect allows for improving the spatial understanding of the active layer dynamics with a minimal environmental disturbance in comparison to boreholes. It allowed detecting high-temporal resolution changes on freezing and thawing along the transect, providing new insight also into the potential geomorphic dynamics and its regime, for example, for processes such as cryoturbation or solifluction.

**Acknowledgments**

The research was funded by the Fundação para a Ciência e a Tecnologia under project PERMANTAR-2 (Permafrost and Climate Change in the Maritime Antarctic – FCT – PTDC/AAC-CLI/098885/2008) and the Portuguese Polar Program (PROPOLAR-FCT). We thank the Spanish Antarctic Station Gabriel de Castilla, the BIO Hespérides personnel for logistical

support, and the continued support of the Spanish Polar Committee for the research in Deception Island. Gabriel Goyanes, Vanessa Baptista, José Miguel Cardoso, Ana David, Alice Ferreira and Mário Neves are thanked for the support in the maintenance of the A-ERT system. This publication is supported by the FCT- project UID/GEO/50019/2019 - Instituto Dom Luiz.

**Data availability**

The time-lapse ERT and borehole temperature data can be obtained on request from the authors.



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





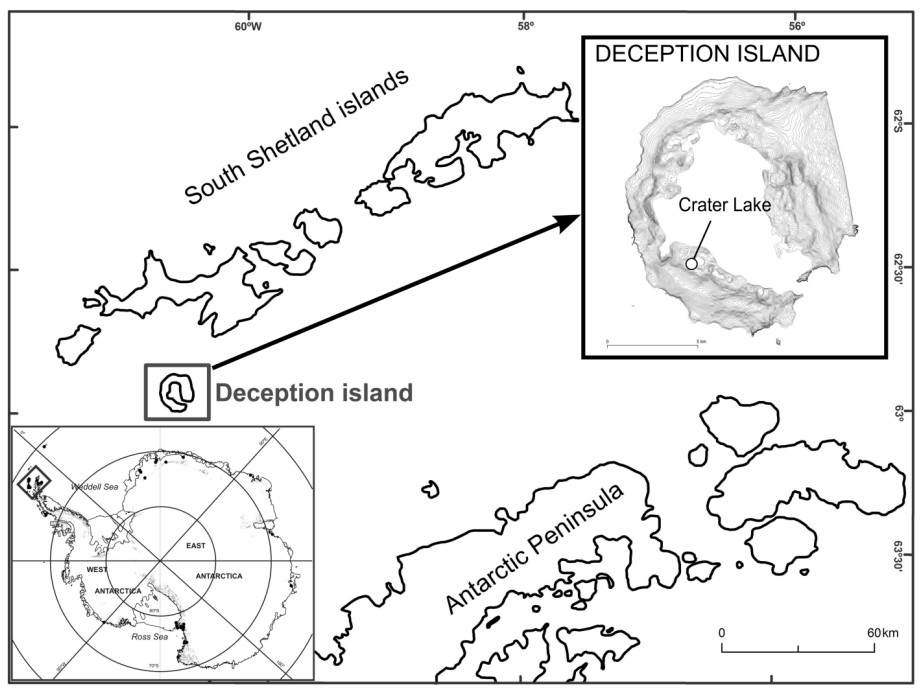

**Figure 1. Location of Deception Island and Crater Lake CALM-S site in Antarctica.**

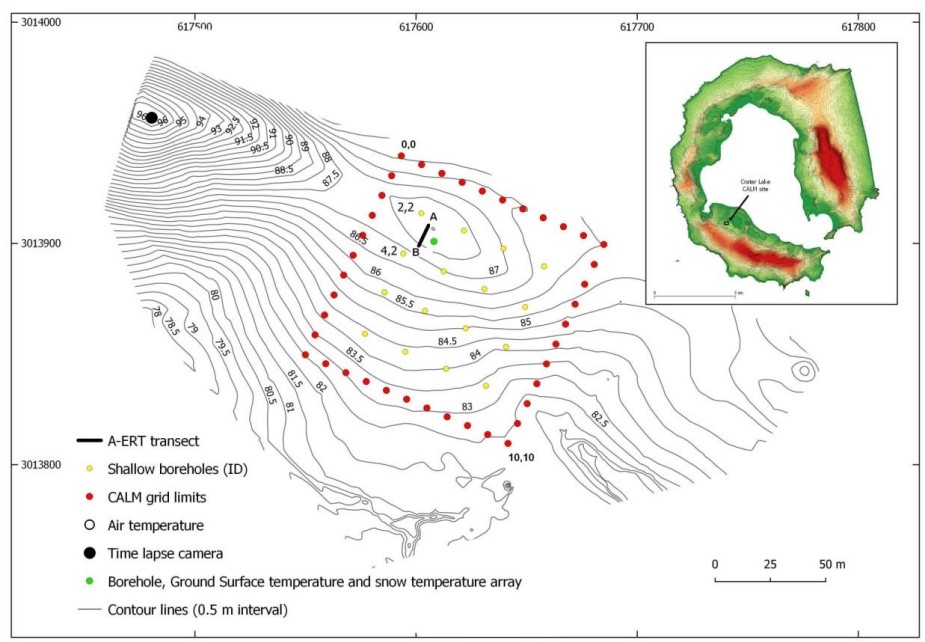

**Figure 2. Environmental monitoring setup at the Crater Lake CALM-S site.**




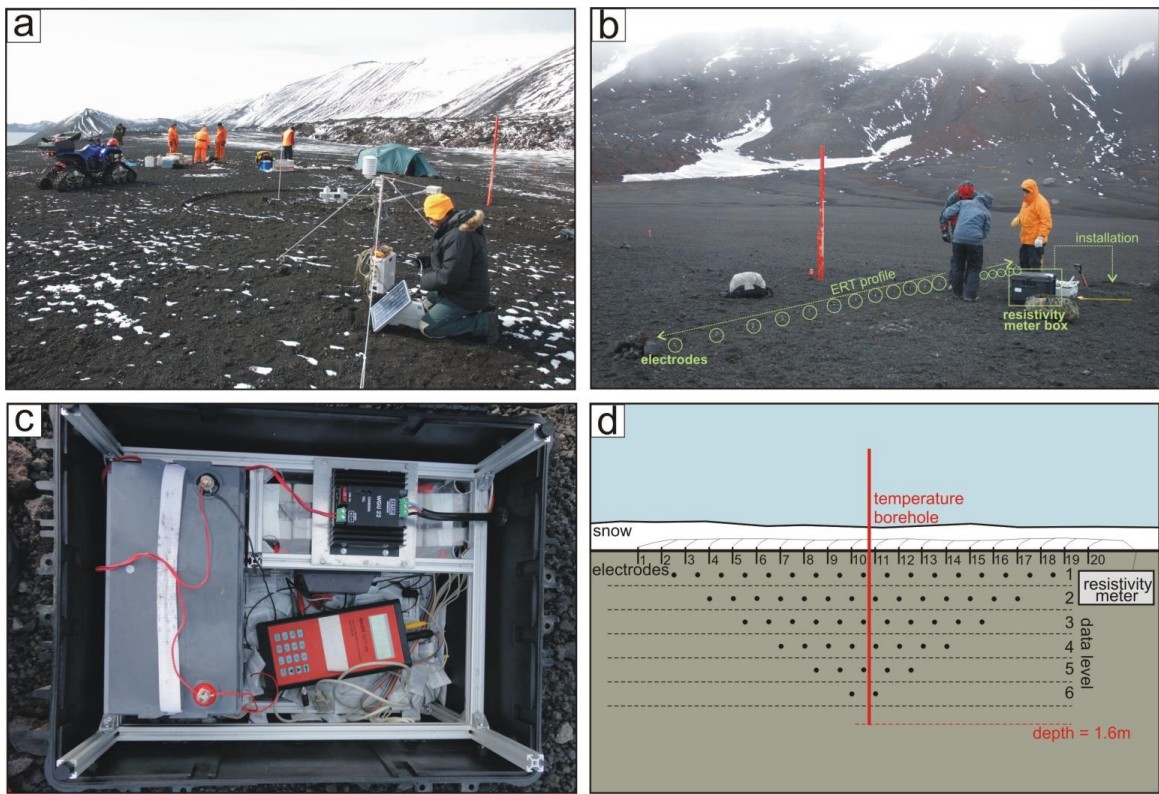

Figure 3. a) Overview of the CALM-S site, b) A-ERT monitoring system installation at CALM-S site; electrodes are buried in the ground and are connected to the resistivity meter box by buried cables, c) Resistivity meter box; the 4POINTLIGHT_10W instrument is connected to a solar panel-driven battery and multi-electrodes connectors . d) A schematic display of the measured resistivities (Pseudo Section) in the CALM-S site using a Wenner electrode configuration.





**Figure 4. (a)** Snow thickness, air and soil surface temperatures variability during the A-ERT data acquisition in 2010. **(b)** Borehole temperature plotted for the sensors installed at the node 3,3 ($S_{3,3}$) covering the investigation depth of the ERT transect. **(c)** Shallow borehole temperatures plotted for the sensors installed at nodes 2,2 and 4,2 at the base of the active layer. The dashed lines mark the selected dates for the ERT inversion analysis shown in Figure 7.



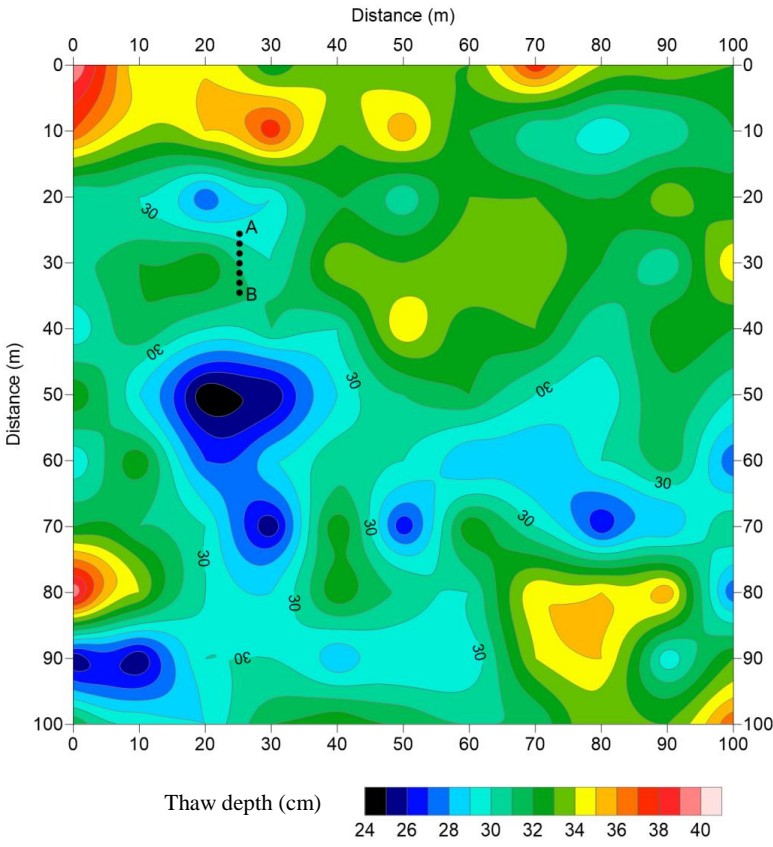

**Figure 5. Spatial distribution of thaw depth measured at the grid nodes across the study area in January 2010. The location of the A-ERT transect is delineated with the black dashed line (A-B).**



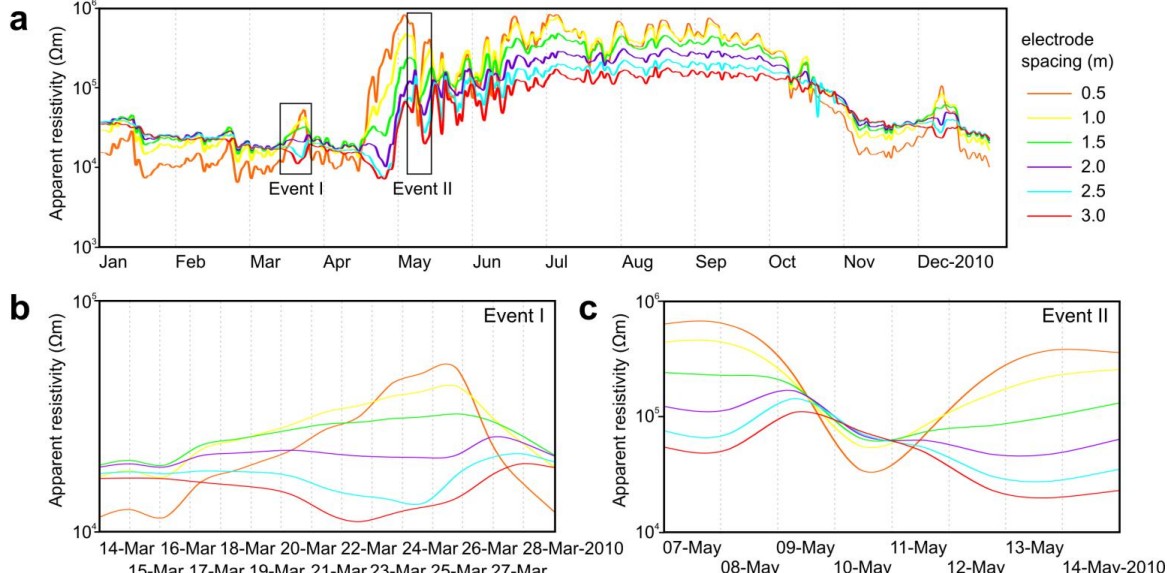

**Figure 6. (a) Mean apparent resistivity data of the A-ERT profile during 2010 for different electrode spacing on a daily scale. (b,c) Mean apparent resistivity data on the scale of individual events: (b) brief surficial refreezing event from 14–28 March 2010, (c) brief surficial thawing event from 7–14 May 2010.**



**Figure 7. (Left) inverted resistivity tomograms of 12 monthly spaced A-ERT datasets between January and December 2010, based on data measured on the 28th of each month; (Right) relative resistivity changes based on the first ERT dataset referred to January.**





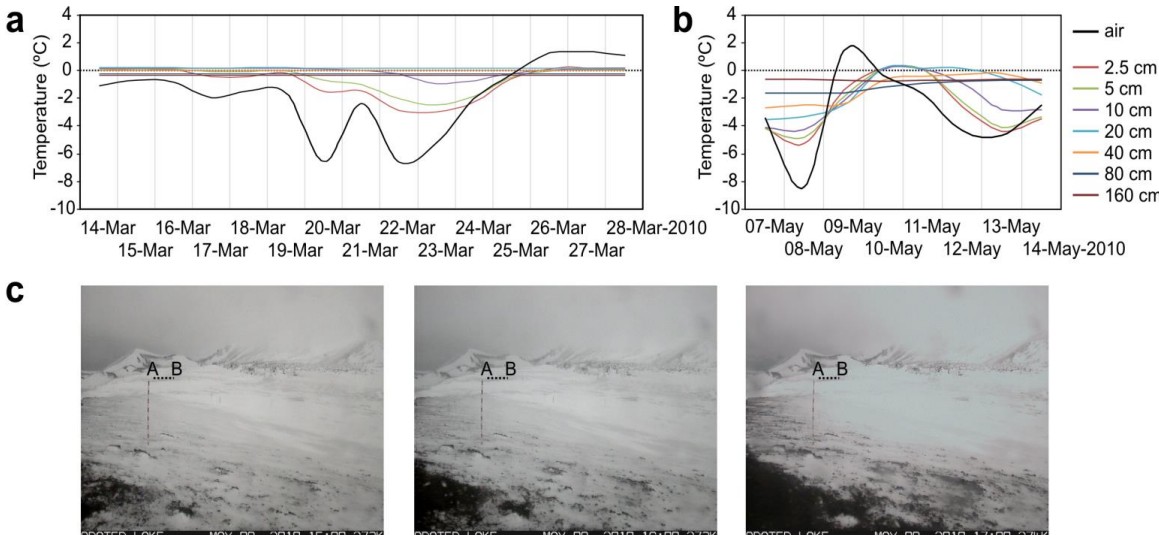

**Figure 8. Air and ground temperature fluctuations in the event scale (a) Event (I): brief surficial refreezing event from 14–28 March 2010, (b) Event (II): brief surficial thawing event from 7–14 May 2010. (c) Time-lapse camera photos at 11:00, 12:00 and 13:00 on May 9.**





**Figure 9. Relative resistivity changes of daily spaced A-ERT datasets in the event scale (a) Event (I): brief surficial refreezing event from 14–28 March 2010, (b) Event (II): brief surficial thawing event from 7–14 May 2010.**



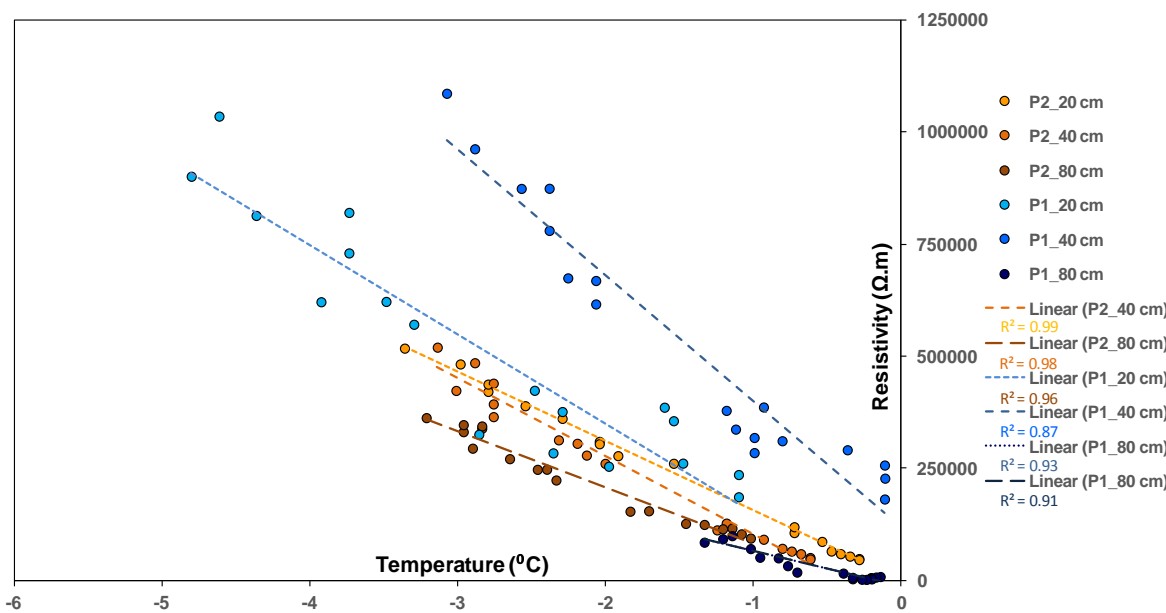

**Figure 10. Resistivity values at the borehole location against borehole temperatures in S$_{3,3}$ during the seasonal active layer freezing in April/May (P1) and thawing in October (P2).**

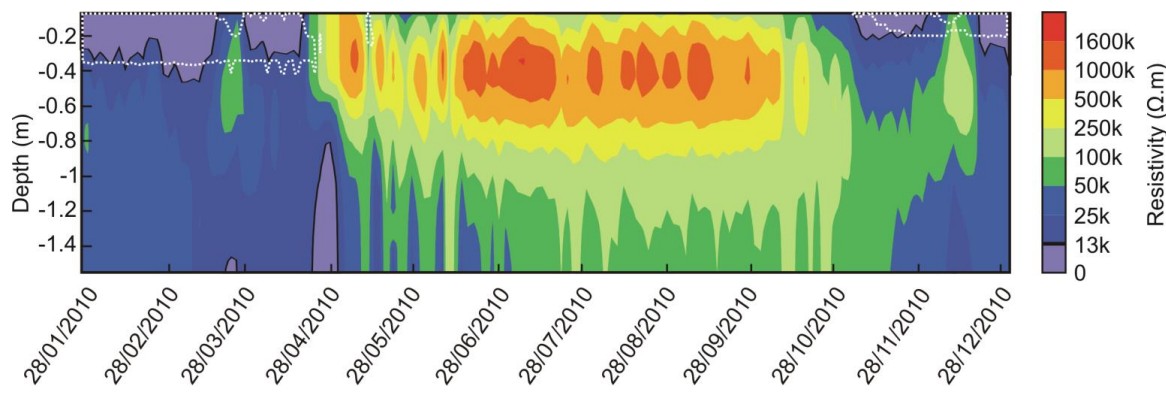

**Figure 11. Evaluation of the temporal resistivity variability in virtual borehole S$_{3,3}$ inferred from inverted A-ERT data for the period January 2010 to December 2010. The black line delineates the cut-off value of 13 k Ohm.m and the white dashed line shows the zero degree isotherm from the borehole temperatures at S$_{3,3}$.**

