# Peer review of "Detailed detection of active layer freeze-thaw dynamics using quasicontinuous electrical resistivity tomography (Deception Island, Antarctica)"

_The Cryosphere, 2019_

## Referee Comment (RC1) · Anonymous Referee #1 · 17 Jun 2019

In this study, presented by M. Farzamian et al. a quasi autonomous electrical resistivity tomography was applied in Crater Lake research site, Antarctica. The study shows the potential to describe fast changes in the active layer of a remote permafrost dominated region with a relatively easy measurement set up. The overall quality of the paper is good, I have only minor points to add or change:

1) p. 3., l. 30 : Please reconsider citing order, this seems a bit odd! 2) p.5 : Are the snow fields interpolated from the mini loggers? If so, how? Could Machine Learning be a method in addition to the camera? 3) p.7, l.11: How does RES2DINV robust inversion

better resolve the contrasts? Please elaborate! 4) p.16., l.11: Can these processes be linked to a mass bilance of the active layer? This would help to understand the hydrological processes taking place in this system. 5) Discussion and outlook: How can this knowledge be transferred into new research? Would a new modelling technique enhance us to get more information from this data? Think about physical Agent-based models (Mewes & Schumann 2018: IPA (v1): a framework for agent-based modelling of soil water movement): The continuous ERT data could be the basis for dynamic models like the one presented by Mewes and Schumann.

---

## Referee Comment (RC2) · Anonymous Referee #2 · 11 Oct 2019

The manuscript "Detailed detection of fast changes in the active layer using quasicontinuous electrical resistivity tomography (Deception Island, Antarctica)" by Farzamian et al. presents automatic ERT measurements covering an entire yearly cycle at very high time resolution at a remote location in Antarctica. Such measurements are extremely rare, and the results are of high interest for the audience of TC. The manuscript, however, could be significantly improved by better focusing on the obvious question "can such a system provide added value compared to traditional techniques". I recommend reviewing the content in this light, which should make it possible to identify superfluous

parts and shorten the manuscript to some extent.

Major comments:

1. The authors show that the ERT setup facilitates estimating/measuring ground temperatures, and the results are indeed impressive. However, ground temperature is exactly the physical variable characterizing the thermal state of permafrost that we can measure very well already. Playing devil's advocate, one could bring forward that a bunch of reliable and inexpensive temperature loggers could easily cover the spatial and temporal scales that the ERT was set up for (at better accuracy and probably lower cost). This is especially true since point-by-point calibration seems necessary (Fig. 10) to convert resistivity to temperature. Ice and water contents, on the other hand, are really hard to measure. The authors mention the transient layer as an example for changing ice contents the Introduction (P. 3, l. 14, see comment below), but no results are presented. If anything (semi-)quantitative regarding water and ice contents can be extracted from the measurements, that would make the manuscript much stronger. I do not question the selected setup or the results presented, but the authors should provide the reader with a better sense of direction where they are going with their research, and how far they have come with the presented results in this process.

2. The fast temperature changes ("events") are interesting (and once again, it is impressive that the ERT can pick them up). But most of the manuscript describes the general evolution over a year which is even more important than fast changes for the above-mentioned question "does such a system provide added value compared to traditional techniques". I therefore recommend changing the title, not mentioning "fast changes", to better describe the (adequate) content of the manuscript.

3. In the Discussion, I am missing a reflection on the points that the ERT system could indeed beat traditional (temperature) measurement techniques. In addition to water and ice contents (see above), this could in particular be temperature monitoring in deeper layers, for which expensive boreholes are needed. Furthermore, spatially resolved (in lateral direction) measurements could be achieved by ERT, and some pattern is indeed visible in Figs. 7 and 9, but this is not discussed in much detail, except for the short paragraph on page 12. This could be discussed in more detail in the Discussion. Another application could be monitoring of changing salinities. The authors should discuss under which circumstances these different variables could be estimated/evaluated in permafrost settings.

Minor Comments:

P2

-l. 2-8: Please break this sentence up in several.

-l. 13: "indicates that our system set-up can successfully map spatiotemporal thaw depth variability" How quantitative does it become with respect to AL? Is "map" which means that spatial differences can be resolved the correct word to use here?

-l. 20: please describe the advantages/added value compared to having a borehole?

P3

-l. 14ff: does the presented method shed any new light on this issue?

P4

-l. 21: Isn't the site rather untypical for "conditions found in Antarctica", when considering all of Antarctica?

P5

-l. 22: please make clear that the "nodes" refer to active layer measurements, this could be confused with the actual ERT setup.

P. 7

-l. 9: delete "e.g."

P16

L. 17: Is this realistic, considering the spatial resolution which would be required for a process like cryoturbation (for example at a mudboil)? What is the spatial resolution in the horizontal direction, that can be resolved by the system? Is it conceivable to use such a system with even finer spacing (e.g. 10cm) and still obtain good results?

---

## Author Comment (AC1) · 5 Dec 2019

Dear editor and reviewers,

We would like to thank you sincerely for your constructive comments of our manuscript. We have revised the manuscript according to your suggestions and comments. Please find below a point-by-point response. We hope that the revised version of the manuscript properly addresses your concerns. Please note that we provide our answers in blue below each of the reviewer's comment.

Sincerely,

Mohammad Farzamian on behalf of all authors

**Author response to reviewer comment**

**Referee #1:**

In this study, presented by M. Farzamian et al. a quasi-autonomous electrical resistivity tomography was applied in Crater Lake research site, Antarctica. The study shows the potential to describe fast changes in the active layer of a remote permafrost dominated region with a relatively easy measurement set up. The overall quality of the paper is good, I have only minor points to add or change:

We would like to thank Anonymous Referee #1 for evaluating our manuscript. We highly appreciate the overall positive comments.

**1) p. 3., l. 30: Please reconsider citing order, this seems a bit odd!**

We corrected it accordingly.

**Old Version (Page 3. Line 30)**

(e.g. Ramos et al., 2008; Vieira et al., 2010; Bockheim et al., 2013; Melo et al., 2012; Goyanes et al., 2014; Ramos et al., 2017)

**New version**

(e.g. Ramos et al., 2008; Vieira et al., 2010; Melo et al., 2012; Bockheim et al., 2013; Goyanes et al., 2014; Ramos et al., 2017)

**2) p.5: Are the snow fields interpolated from the mini loggers? If so, how? Could Machine Learning be a method in addition to the camera?**

The temperature miniloggers allow for estimating the snow thickness at two points only, however, the images from the time-lapse camera allow for detecting the snow cover position. In this study, we used a manual analysis for determining the timing and location of the snow cover (an automated algorithm for detecting snow occurrence from time-lapse camera pictures is available as well, but was not used in this case due to the comparatively short time-series). No machine learning is necessary in this case, since the contrast between the snow and the ash surface is really sharp. No changes of the text were made in relation to this comment.

**3) p.7, l.11: How does RES2DINV robust inversion better resolve the contrasts? Please elaborate!**

The conventional smoothness-constrained least squares method, which is widely used in inversion algorithms, attempts to minimize the square of the changes in the model resistivity values. Therefore, this method produces a model with a smooth variation in the resistivity, values which is suitable where subsurface resistivity also changes in a smooth manner. However, if the subsurface has sharp boundaries, such as the unfrozen/frozen soil interface (large contrast in resistivity), this conventional method tends to smear the boundaries. In this regard, the so-called robust model works better as the objective function attempts to minimize the absolute changes in the resistivity values, which produces models with sharp interfaces between different regions with different resistivity values, but within each region the resistivity value is almost constant. For better explanation we have revised the manuscript as follows:

**Old Version (Page 7. Line 9)**

In the next step, the apparent resistivity datasets were inverted using e.g. the commercially available software RES2DINV. The robust inversion option in RES2DINV, as well as a mesh refinement to half of the electrode spacing, was applied to better resolve the expected strong resistivity contrasts between unfrozen and frozen subsurface materials.

**New version**

In the next step, the apparent resistivity datasets were inverted using the commercially available software RES2DINV. The robust inversion option in RES2DINV, as well as a mesh refinement to half of the electrode spacing, was applied to better resolve the expected strong resistivity contrasts between unfrozen and frozen subsurface materials. The objective function used in the robust inversion algorithm attempts to minimize the absolute changes in the resistivity values which produces models with sharp interfaces between different regions with different resistivity values (Loke, 2002).

4) p.16., l.11: Can these processes be linked to a mass bilance of the active layer? This would help to understand the hydrological processes taking place in this system.

This is a very interesting suggestion, but goes quite beyond the scope of the present paper, In a more complex integrated approach (hydrogeophysical modelling approach, as mentioned in the preceding paragraph) the A-ERT models could in principle be used to assess the mass balance of the system. First attempts to couple resistivity data with hydro-thermal modelling were e.g. made in a recent paper by Jafarov et al. (2019), but using only synthetic data, Tomaskovicova (2018) and on smaller time scales by Scherler et al. (2010). However, in all cases this would require supplementary information regarding the hydrological setting of the study site, in-situ data (e.g. soil moisture) for model calibration as well as a proper uncertainty assessment of the geophysical models (e.g. data quality, inversion artifacts) with respect to the available in-situ data. At the moment, the monitoring station at Deception Island does not provide this kind of additional data.

**References:**

Jafarov, E. E., Harp, D. R., Coon, E. T., Dafflon, B., Tran, A. P., Atchley, A. L., Wilson, C. J., and Lin, Y. (2019): Estimation of soil properties by coupled inversion of electrical resistance, temperature, and moisture content data, The Cryosphere Discuss., https://doi.org/10.5194/tc-2019-91, in review.

Scherler, M., Hauck, C., Hoelzle, M., Stähli, M. and Völksch, I. (2010): Meltwater infiltration into the frozen active layer at an alpine permafrost site. Permafrost and Periglacial Processes 21: 325–334, DOI: 10.1002/ppp.694.

Tomaskovicova, S. (2018). *Coupled thermo-geophysical inversion for permafrost monitoring*. PhD thesis Technical University of Denmark, Department of Civil Engineering. BYGDTU. Rapport, No. R-387

5) Discussion and outlook: How can this knowledge be transferred into new research? Would a new modelling technique enhance us to get more information from this data? Think about physical Agent-based models (Mewes & Schumann 2018: IPA (v1): a framework for agent-based modelling of soil water movement): The continuous ERT data could be the basis for dynamic models like the one presented by Mewes and Schumann.

We have revised the last two paragraphs of the manuscript by including several potential areas of future extension and application of the A-ERT approach in general and in our study area:

**Old Version (Page 16. Line 4)**

[revised manuscript text omitted]

**Referee #2:**

The manuscript "Detailed detection of fast changes in the active layer using quasi continuous electrical resistivity tomography (Deception Island, Antarctica)" by Farzamian et al. presents automatic ERT measurements covering an entire yearly cycle at very high time resolution at a remote location in Antarctica. Such measurements are extremely rare, and the results are of high interest for the audience of TC. The manuscript, however, could be significantly improved by better focusing on the obvious question "can such a system provide added value compared to traditional techniques". I recommend reviewing the content in this light, which should make it possible to identify superfluous parts and shorten the manuscript to some extent.

We would first like to thank Anonymous Referee #2 for evaluating our manuscript. We appreciate the overall positive comment that highlight the potential impact of our research. We have now revised the manuscript in this light to further highlight the potential of the A-ERT system compared to traditional methods. Please see our detailed answers to the various individual comments below.

**Major comments:**

1. The authors show that the ERT setup facilitates estimating/measuring ground temperatures, and the results are indeed impressive. However, ground temperature is exactly the physical variable characterizing the thermal state of permafrost that we can measure very well already. Playing devil's advocate, one could bring forward that a bunch of reliable and inexpensive temperature loggers could easily cover the spatial and temporal scales that the ERT was set up for (at better accuracy and probably lower cost). This is especially true since point-by-point calibration seems necessary (Fig. 10) to convert resistivity to temperature. Ice and water contents, on the other hand, are really hard to measure. The authors mention the transient layer as an example for changing ice contents the Introduction (P. 3, 1. 14, see comment below), but no results are presented. If anything (semi-)quantitative regarding water and ice contents can be extracted from the measurements, that would make the manuscript much stronger. I do not question the selected setup or the results presented, but the authors should provide the reader with a better sense of direction where they are going with their research, and how far they have come with the presented results in this process.

It is correct that we showed how ERT data could estimate ground temperature due to the strong resistivitytemperature correlation, however, this is not the main aim of this study as described in the manuscript. With this study we aim to I) evaluate the feasibility of installing and running autonomous ERT monitoring stations in remote and extreme environments such as Antarctica, II) to monitor subsurface freezing and thawing processes on a daily and seasonal basis and mapping the spatial and temporal variability of thaw depth and III) to study the impact of short-lived extreme meteorological events on active layer dynamics. Validating and illustrating the performance of the A-ERT system by using near-surface temperature dynamics does therefore not mean that we propose to use the system only for this target – as the reviewer remarked correctly, for this objective, shallow boreholes with temperature measurements are equally well suited, if they can be deployed. However, being able to predict ground temperatures (also at much larger depths) using a non-invasive method and hence, low environmental impact method, will be indeed very valuable especially in Antarctica, where the ecosystem is very sensitive to invasive techniques. In addition, drilling (not the temperature loggers) is very expensive in Antarctica, while A-ERT set-up is a comparatively inexpensive method with flexible set-up to investigate different depths. Moreover, using an A-ERT system, we could detect freeze-thaw process of active layer in very high temporal and spatial resolution which cannot be easily assessed by temperature loggers as phase changes are not explicitly seen. Having e.g. a series of 20 m deep boreholes in frozen soil/rocky terrain over a transect of several 100 metres (to cover its heterogeneity) is quite impossible to achieve in Antarctica, whereas an A-ERT installation of this size would only require longer cables than the one shown in our study.

We have revised the text in this light, please see also our detailed answer to the comment #3 in this regard.

The feasibility of quantitative water/ice content estimation from A-ERT data was shown in several studies; however, this requires a combination of data processing techniques, petrophysical models and supporting information (e.g. Fortier et al., 2008; Hauck et al., 2011; Grimm and Stillman, 2015; Dafflon et al., 2016) to provide a proper water content estimation. In addition, to access a reliable estimate water/ice contents from resistivity alone, the type of electric conduction must be analysed. Duvillard et al. (2018) showed that for comparatively dry soils with low salinity, surface conduction is the dominant process as opposed to electrolytic conduction, which is usually assumed to calculate water contents from resistivity values (e.g. by using Archie's Law). In the context of the volcanic material at Deception Island, the link between pore water resistivity and measured bulk resistivity would have to be made by laboratory measurements, which has not been done in the present study. Consequently, an explicit link between monitored resistivity values and ice/water content would be possible and is desirable, but to do this in a reliable manner would go beyond the scope of the present study. In this regard, we revised the "Conclusion and Outlook" section to better address the potential of A-ERT in water/ice content estimation.

**Old Version (Page 16. Line 4)**

The consistency of our full year results with previous studies in more easily accessible alpine and polar regions (e.g. Hilbich et al., 2011; Supper et al., 2014; Keuschnig et al., 2017; Tomaskovicova, 2017, Oldenborger and LeBlanc, 2018) suggests that the detailed studies of the Alps can be transferred to set-ups in very remote environments, which would allow for integrative process studies as well as coupled modeling of A-ERT data with existing water content and temperature monitoring system in Antarctica....

**New version**

The consistency of our full year results with previous studies in more easily accessible alpine and polar regions (e.g. Hilbich et al., 2011; Supper et al., 2014; Keuschnig et al., 2017; Tomaskovicova, 2017, Oldenborger and

LeBlanc, 2018) suggests that the detailed studies of the Alps can be transferred to set-ups in very remote environments, which would allow for integrative process studies as well as coupled modeling of A-ERT data with existing water content and temperature monitoring system in Antarctica. Examples of such studies are combination of data processing techniques, petrophysical models and supporting information to estimate unfrozen water content from electrical resistivity data (e.g. Hauck, 2002; Fortier et al., 2008; Grimm and Stillman, 2015; Dafflon et al., 2016) or combining electrical resistivity data with seismic refraction data in a joint petrophysical model to estimate ice and water content (e.g. Hauck et al., 2011). Such analyses also provide a tool to monitor the transient layer and study the impact of fast-changing meteorological conditions and frequent freeze-thaw process on soil behavior at the permafrost table. However, in the context of the volcanic material at Deception Island, the link between pore water resistivity and measured bulk resistivity should be assessed by laboratory measurements prior to performing a quantitative investigation on soil ice/water content. In addition, the type of the electric conduction needs to be investigated as in dry soils with low salinity, the surface conduction is the dominant process (Duvillard et al., 2018) as opposed to electrolytic conduction which is usually assumed to calculate water contents from resistivity values.

Please see our answer to comment 1. We revised the manuscript in this light and reflected the advantage of the non-invasive A-ERT system compared to the boreholes. This includes the revision of the abstract, introduction, and discussion with the following details:

**Abstract: we revised the last paragraph of the abstract as follows:**

**Old Version (Page 2. Line 22)**

Based on this first complete year-round A-ERT monitoring data set in Deception Island, we believe that this system shows high potential for autonomous applications in remote and harsh polar environments such as Antarctica.

**New version**

Based on this first complete year-round A-ERT monitoring data set in Deception Island, we believe that this system shows high potential for autonomous applications in remote and harsh polar environments such as Antarctica. In addition, the monitoring system can be used with larger electrode spacing to investigate greater depths, providing adequate monitoring at sites and depths where boreholes are very costly and the ecosystem is very sensitive to invasive techniques. Further applications may be the estimation of ice/water contents through petrophysical models or the calibration/validation of heat transfer models between active layer and permafrost.

**Introduction: we revised two paragraphs of the introduction as follows:**

**Old Version (Page 4. Line 2)**

... In addition, being an invasive technique, the drilling of boreholes disturbs the subsurface and is not feasible to conduct over large areas, especially in environmentally sensitive ecosystems such as the Antarctic.

**New version**

... In addition, being an invasive technique, the drilling of boreholes disturbs the subsurface and is not feasible to conduct over large areas, especially in environmentally sensitive ecosystems such as the Antarctic. Moreover, drilling boreholes to monitor temperature in deeper layers are very expensive in Antarctica, which further limits the application of boreholes in deep investigations and in areas with very heterogeneous ground conditions.

**Old Version (Page 4. Line 10)**

.... Due to the large contrast between the resistivity of ice and water, the method has become popular in permafrost investigation to distinguish between frozen and unfrozen soil.

**New version**

.... Due to the large contrast between the resistivity of ice and water, the method has become popular in permafrost investigation to distinguish between frozen and unfrozen soil and thus to monitor the active layer dynamics including freezing, thawing, water infiltration and refreezing processes in a spatial context, which is sometimes very difficult to assess with only temperature boreholes. This technique is also being widely used to provide non-invasive estimates of spatiotemporal unfrozen water content distribution due to the strong dependence of electrolytic conduction on the phase change of water to ice in earth materials (e.g. Hauck, 2002).

**Discussion: we revised a paragraph of the discussion as follows:**

**Old Version (Page 14. Line 31)**

.... Hence, the ground temperature tomogram (Fig. 4b) shows a constant thaw depth of 40 cm in the first three months...

**New version**

... Hence, the ground temperature tomogram (Fig. 4b) shows a constant thaw depth of 40 cm during the first three months. These results reveal that our A-ERT set-up allows for accurate characterization of the active layer freeze-thaw process, with a spatial resolution that can usually not be achieved with temperature sensors, except for a very dense sensor setup...

The lateral resistivity variations seen in Figures 7 and 9 were already discussed in the results section. To make the text clearer and more consistent in this regard, we have reformulated the manuscript and included a new paragraph about the lateral resistivity changes along the A-ERT transect in the "discussion" section as follows:

The resistivities of both, active layer and permafrost zones, indicate only a slight lateral change along the transect. This can be explained by the spatial homogeneity of the study area, as well as by the small size of the A-ERT transect, which is smaller than for other A-ERT studies where stronger lateral variations along the ERT transects are usually more evident (i.e. Hilbich et al., 2011; Supper et al., 2014; Keuschnig et al., 2017). However, larger lateral resistivity changes are visible during the extreme short-lived meteorological events. An example of such lateral changes is very evident during Event (II) shown in Figure 9b. The obtained resistivity models during this event suggest the propagation of the thawing process from the left (A) to the right (B) on May 9 and 10 (i.e. the active layer resistivity decreased from the left to the right) and then refreezing from the same direction on May 11. Because the left side of the A-ERT transect is closer to the interfluve and is more wind and sun exposed, subsurface thaw and snowmelt are expected to take place from left to right along the transect orientation after the initial air temperature rise on May 9. Similarly, active layer refreezing starts from the same direction (left to right) when the air cools down again on May 11. On the seasonal basis, a similar lateral resistivity variation is visible in Figure 7. During the freezing season, the resistivity of the active layer is higher in the left side due to the enhanced cooling of the active layer in this part of the profile. Similarly, the resistivity of the active layer decreases from the left to the right during active layer warming (i.e. September 2010) and thawing (i.e. October 2010).

The ashes at Deception Island are recent (a few hundred years) and pervious unpublished studies in this site indicate a non-saline soil. Our geophysical investigations across the site also show very resistive soil (active layer) which also suggest that the soil is not saline.

**Minor Comments:**

P2 -1. 2-8: Please break this sentence up in several.

We revised it accordingly as follows:

**Old Version (Page 2. Line 2)**

Climate induced warming of permafrost soils is a global phenomenon, with regional and site-specific variations, which are not fully understood. In this context, a 2D automated electrical resistivity tomography (A-ERT) system was installed for the first time in Antarctica at Deception Island, associated to the existing Crater Lake site of the Circumpolar Active Layer Monitoring Network (CALM-S) I) to evaluate the feasibility of installing and running autonomous ERT monitoring stations in remote and extreme environments such as Antarctica, II) to monitor subsurface freezing and thawing processes on a daily and seasonal basis and to map the spatial and temporal variability of thaw depth, and III) to study the impact of short-lived extreme meteorological events on active layer dynamics.

**New version**

Climate induced warming of permafrost soils is a global phenomenon, with regional and site-specific variations, which are not fully understood. In this context, a 2D automated electrical resistivity tomography (A-ERT) system was installed for the first time in Antarctica at Deception Island, associated to the existing Crater Lake site of the Circumpolar Active Layer Monitoring Network (CALM-S). This set-up aims to I) monitor subsurface freezing and thawing processes on a daily and seasonal basis and to map the spatial and temporal variability of thaw depth, and to II) study the impact of short-lived extreme meteorological events on active layer dynamics. In addition, the feasibility of installing and running autonomous ERT monitoring stations in remote and extreme environments such as Antarctica was evaluated for the first time.

-l. 13: "indicates that our system set-up can successfully map spatiotemporal thaw depth variability" How quantitative does it become with respect to AL? Is "map" which means that spatial differences can be resolved the correct word to use here?

We have revised this paragraph and we use now the term "resolve". In addition, according to the electrode spacing and array configuration, we expect a resolution of 20-30 cm.

-1. 20: please describe the advantages/added value compared to having a borehole?

**We revised this section as follow:**

**Old Version (Page 2. Line 22)**

Based on this first complete year-round A-ERT monitoring data set in Deception Island, we believe that this system shows high potential for autonomous applications in remote and harsh polar environments such as Antarctica.

**New version**

Based on this first complete year-round A-ERT monitoring data set in Deception Island, we believe that this system shows high potential for autonomous applications in remote and harsh polar environments such as Antarctica. In addition, the monitoring system can be used with larger electrode spacing to investigate greater depths, providing adequate monitoring at sites and depths where boreholes are very costly and the ecosystem is very sensitive to invasive techniques. Further applications may be the estimation of ice/water contents through petrophysical models or the calibration/validation of heat transfer models between active layer and permafrost.

**P3 -l. 14ff: does the presented method shed any new light on this issue?**

In fact, this layer was resolved in our resistivity section in Figure 11. The higher amount of ice in this layer increases the resistivity which could be detected in the inverted resistivity models. We have revised the "discussion" section and added the following explanation in this regard:

**Old Version (Page 14. Line 31)**

... Hence, the ground temperature tomogram (Fig. 4b) shows a constant thaw depth of 40 cm in the first three months....

**New version**

... Hence, the ground temperature tomogram (Fig. 4b) shows a constant thaw depth of 40 cm during the first three months. These results reveal that our A-ERT set-up allows for accurate characterization of the active layer freeze-thaw process, with a spatial resolution that can usually not be achieved with temperature sensors, except for a very dense sensor setup. In addition, the spatiotemporal resistivity variations show that the resistivity values are greatest in winter and around the permafrost table at depths around 40 cm (see Fig. 11). This is due to the repeated thawing and refreezing processes of water infiltrating from snow/rain that accumulated on top of the permafrost table (cf. the transition zone, Shur et al., 2005) which forms an ice-rich layer and increases its resistivity. Plotting the resistivity values at the borehole location against borehole temperatures in  $S_{3,3}$  (Fig. 10)

also shows remarkably greater resistivity values during the active layer freezing at a depth of 40 cm revealing the fact that the A-ERT data could be used to study the transition zone in the study area.

**P4 -1. 21: Isn't the site rather untypical for "conditions found in Antarctica", when considering all of Antarctica?**

The site is not typical for soil conditions in Antarctica, since it is a volcanic setting. However, the sentence relates to logistics and for these, Deception Island is similar to most other areas in the Antarctic Peninsula and also other Antarctic stations without year-round maintenance. All available bases on Deception Island are only summer operated.

P5 -1. 22: please make clear that the "nodes" refer to active layer measurements, this could be confused with the actual ERT setup.

We have revised this section accordingly as follow:

**Old Version (Page 5. Line 22)**

The Crater Lake CALM-S site consists of a  $100 \times 100$  m grid with 121 nodes spaced at 10 m intervals and was installed in January 2006 (Fig. 2) with several upgrades since then. The site includes monitoring of air temperature, permafrost and the active layer in boreholes, snow thickness and once per year, thaw depth is measured manually by mechanical probing during the summer (Ramos et al., 2017).

**New version**

The Crater Lake CALM-S site consists of a  $100 \times 100$  m grid and was installed in January 2006 (Fig. 2) with several upgrades since then. The site includes monitoring of air temperature, permafrost and the active layer in boreholes, and snow thickness. Thaw depth is measured manually once per year during summer at 121 nodes spaced at 10 m intervals by mechanical probing (Ramos et al., 2017).

**P. 7 -l. 9: delete "e.g."**

We corrected it accordingly.

P16 L. 17: Is this realistic, considering the spatial resolution which would be required for a process like cryoturbation (for example at a mudboil)? What is the spatial resolution in the horizontal direction, that can be

resolved by the system? Is it conceivable to use such a system with even finer spacing (e.g. 10cm) and still obtain good results?

The spatial resolution of the A-ERT system is a function of the selected electrode spacing, array configuration and subsurface properties. Regarding the small spacing of 50 cm in our set-up we expect lateral resolution of 20-30 cm (Figure 3 d), which allows for accurate characterization of the freeze-thaw process with a resolution that, with temperature sensors, would only be possible with a very dense setup. This allows for detecting changes, potentially, even at the large mudboil scale, and hence detect changes between center and border frost penetration. However in mudboils, close to the surface, micro-topography would need careful modelling. In the study area, no mudboils occur and in fact, ice segregation is limited by the coarse and porous nature of the lappilli. In sloping terrain, the A-ERT system could be used to detect water movement over the frozen soil layer and the linkage of the process to shallow-debris flow initiation or even shallow active layer detachment slide dynamics.

A smaller electrode spacing of 10 cm could be used to obtain even higher resolution (e.g. 5 cm). Such an electrode spacing is used in "micro" A-ERT monitoring system to study e.g. the soil-plant interactions in laboratory or in field in micro scale (e.g. Boaga et al., 2013). However, this would require smaller electrodes and performing several tests to study feasibility of such a set-up at the site. In addition, performing the same set-up but using the small electrode spacing of 10 cm will yield smaller (~ 1/5) spatial coverage compared to the original configuration, and smaller maximum investigation depth.

**References:**

Boaga, J., Rossi, M., and Cassiani, G.: Monitoring Soil-plant Interactions in an Apple Orchard Using 3-D Electrical Resistivity Tomography, Procedia Environ. Sci., 19, 394–402, https://doi.org/10.1016/j.proenv.2013.06.045, 2013.